



# Ship-borne measurements of $XCO_2$, $XCH_4$, and XCO above the Pacific Ocean and comparison to CAMS atmospheric analyses and S5P/TROPOMI

Marvin Knapp[1], Ralph Kleinschek[1], Frank Hase[2], Anna Agustí-Panareda[3], Antje Inness[3], Jérôme Barré[3], Jochen Landgraf[4], Tobias Borsdorff[4], Stefan Kinne[5], and André Butz[1,6]

[1]Institute of Environmental Physics, Heidelberg University, Germany
[2]Karlsruhe Institute of Technology (KIT), Institute for Meteorology and Climate Research (IMK-ASF), Karlsruhe, Germany
[3]European Centre for Medium-Range Weather Forecasts, Shinfield Park, Reading, RG2 9AX, UK
[4]Netherlands Institute for Space Research, SRON, Utrecht, the Netherlands
[5]Max Planck Institute for Meteorology, Hamburg, Germany
[6]Heidelberg Center for the Environment (HCE), Heidelberg University, Germany

**Correspondence:** Marvin Knapp (marvin.knapp@iup.uni-heidelberg.de)

**Abstract.** Measurements of atmospheric column-averaged dry-air mole fractions of carbon dioxide ($XCO_2$), methane ($XCH_4$), and carbon monoxide (XCO) have been collected across the Pacific ocean during the Measuring Ocean REferences 2 (MORE-2) campaign in June 2019. We deployed a ship-borne variant of the EM27/SUN Fourier Transform Spectrometer (FTS) on board the German research vessel *Sonne* which, during MORE-2, crossed the Pacific ocean from Vancouver, Canada, to Sin-
gapore. Equipped with a specially manufactured fast solar tracker, the FTS operated in direct-sun viewing geometry during the ship cruise reliably delivering solar absorption spectra in the shortwave infrared spectral range (4000 to 11000 $cm^{-1}$). After filtering and bias correcting the dataset, we report on $XCO_2$, $XCH_4$, and XCO measurements for 22 days along a trajectory that largely aligns with 30° N of latitude between 140° W and 120° E of longitude. The dataset has been scaled to the Total Carbon Column Observing Network (TCCON) station in Karlsruhe, Germany, before and after the MORE-2 campaign through side-
by-side measurements. The precision for hourly means of $XCO_2$, $XCH_4$, and XCO during the campaign is found 0.24 ppm, 1.1 ppb, and 0.75 ppb, respectively. Comparing concentration fields analysed by the Copernicus Atmosphere Monitoring Service (CAMS) to our data, we find excellent agreement of 0.52±0.31 ppm for $XCO_2$, 0.9±4.1 ppb for $XCH_4$, and 3.2±3.4 ppb for XCO (mean difference ± standard deviation of differences for entire record). Likewise, we find excellent agreement to within 2.2±6.6 ppb with the XCO observations of the TROPOspheric MOnitoring Instrument (TROPOMI) on the Sentinel-5
Precursor satellite (S5P). The ship-borne measurements are accessible at https://doi.org/10.1594/PANGAEA.917240 (Knapp et al., 2020).

## 1 Introduction

The greenhouse gases carbon dioxide ($CO_2$) and methane ($CH_4$) and the air pollutant carbon monoxide (CO) are the target constituents of a range of currently orbiting and planned Earth observing satellite missions (e.g. Kuze et al., 2009; Eldering

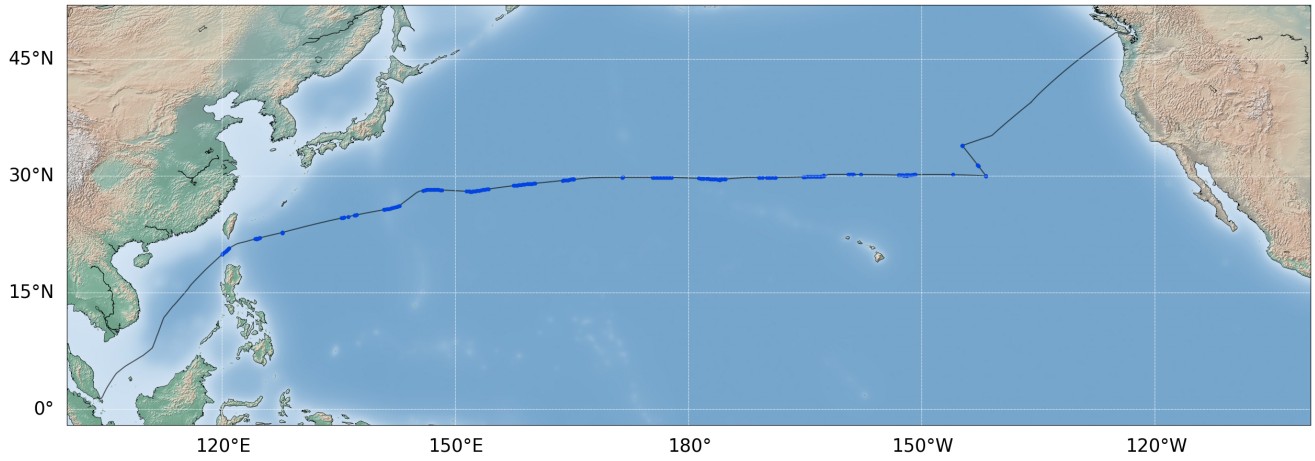

**Figure 1.** Track of the research vessel *Sonne* (grey line) during the MORE-2 campaign starting from Vancouver, Canada, on May 30, 2019, and entering port in Singapore on July 5, 2019. The blue dots are the locations of all quality-assured EM27/SUN measurements. The map is provided by Wessel et al. (2019).

et al., 2017). The latest addition to the fleet of spaceborne sensors is the Sentinel-5 Precursor (S5P) satellite with its TROPO-spheric Monitoring Instrument (TROPOMI) in orbit since October 2017 (Veefkind et al., 2012). TROPOMI retrieves, among other constituents, the column-averaged dry-air mole fractions of $CH_4$ ($XCH_4$) and CO (XCO) from spectra of backscattered sunlight in the shortwave-infrared (SWIR) spectral range (Borsdorff et al., 2017; Hu et al., 2018; Borsdorff et al., 2019; de Gouw et al., 2020). In parallel to the expansion of the fleet of greenhouse gas sensors in orbit, the European Centre for Medium Range Weather Forecasts (ECMWF) operates the Copernicus Atmosphere Monitoring Service (CAMS) on behalf of the European Commision. CAMS assimilates satellite measurements of atmospheric composition to forecast the global $CO_2$, $CH_4$, and CO concentrations at high spatial and temporal resolution using the Integrated Forecasting System (IFS). The ultimate goal is to monitor mitigation of anthropogenic greenhouse gas emissions and air pollution from global to regional scales (Massart et al., 2014, 2016; Inness et al., 2015, 2019; Agustí-Panareda et al., 2019; Janssens-Maenhout et al., 2020). Validation of the $XCO_2$, $XCH_4$, and XCO satellite data mostly relies on ground-based direct-sun spectroscopic observations conducted by the Total Carbon Column Observing Network (TCCON) (Wunch et al., 2011) supplemented by the emerging Collaborative Carbon Column Observing Network (COCCON) (Frey et al., 2019) that measure the column-averaged dry-air mole fractions with similar column-sensitivity as the satellites. Likewise, the CAMS model uses TCCON as an evaluation tool (Agustí-Panareda et al., 2019). Most of the observatories of the TCCON and COCCON are located at continental sites. Thus,





validation of the satellites and models over the oceans is limited to a few island and coastal observatories (in particular Ascension, Reunion, Tenerife, Japan, California). For the satellite retrievals, ocean-land biases (e.g. Basu et al., 2013) can occur since the ocean surface is dark and thus, satellites typically have to resort to glint geometry (e.g. Butz et al., 2013) or retrievals above clouds (e.g. Vidot et al., 2012; Schepers et al., 2016), which impose difficulties different from the typical clear-sky nadir observations above land.

To enable evaluation of satellites and models over the oceans, Klappenbach et al. (2015) developed a ship-borne prototype of the EM27/SUN Fourier Transform Spectrometer (FTS) (Gisi et al., 2011, 2012) which is the instrument used within the COCCON (Frey et al., 2019). The EM27/SUN has proven a reliable instrument for $XCO_2$ and $XCH_4$ measurements in various studies ranging from ad-hoc networks covering a larger region of interest (Hase et al., 2015b; Chen et al., 2016; Toja-Silva et al., 2017; Viatte et al., 2017; Vogel et al., 2019) to mobile deployments (Butz et al., 2017; Luther et al., 2019) for the

quantification of localized $CO_2$ and $CH_4$ sources. The latest variant of the EM27/SUN disposes of a second spectral detector channel that enables XCO measurements simultaneously with observations of $XCO_2$ and $XCH_4$ (Hase et al., 2016). The ship-borne observations by Klappenbach et al. (2015) were conducted on board the research vessel *Polarstern* during a cruise from Cape Town, South Africa, to Bremerhaven, Germany, in March and April 2014. These measurements were used for evaluating $XCO_2$ and $XCH_4$ observations of the Greenhouse Gases Observing Satellite (GOSAT) and for improving the inter-hemispheric

gradient modelled by the IFS for the CAMS $CO_2$ and $CH_4$ analysis and forecasting system (Agusti-Panareda et al., 2017).

   Here, we report on the further developments of the ship-borne EM27/SUN prototype toward routine use as a validation tool over the open oceans. To demonstrate the performance and robustness of the instrumentation and its suitability for satellite and model validation, we deployed the instrument on the German research vessel *Sonne* during the MORE-2 (Measuring Oceanic REferences 2) campaign which led from Vancouver, Canada, to Singapore between May 30, 2019 and July 5, 2019. Figure

1 shows the track of the research vessel through the Pacific Ocean. We report on technical developments (section 2), the data processing chain and data quality assessment (section 3), as well as comparisons to TROPOMI's XCO measurements and CAMS' analyses fields of $XCO_2$, $XCH_4$, and XCO (section 4) over the Pacific ocean. The data collected are publicly available at https://doi.org/10.1594/PANGAEA.917240 for evaluating other datasets, and the ship-borne instrument is recommended for routine deployment on ships.

## 2   Instrumentation


The EM27/SUN is a commercially available FTS which was developed in a cooperation of Bruker Optics and the Karlsruhe Institute of Technology (KIT) (Gisi et al., 2012). The spectrometer has the dimensions $42{\times}27{\times}35\,\mathrm{cm}^3$ and weighs about 25 kg. The EM27/SUN uses a $CaF_2$ beam splitter and a RockSolid™ pendulum interferometer with 2 cube corner mirrors. The maximum optical path difference of 1.8 cm supports a spectral resolution of $0.50\,\mathrm{cm}^{-1}$. After the sunlight passed the

interferometer, a parabolic off-axis mirror focuses it on an InGaAs photodetector with a spectral range of 5500 - 11000 $\mathrm{cm}^{-1}$, further called SWIR-1 channel. Another mirror decouples about 40 % of the beam on a second spectrally extended InGaAs photodetector covering the spectral range of 4000 - 5500 $\mathrm{cm}^{-1}$ (Hase et al., 2016), called SWIR-3 channel. Typical exposure



times are on the order of $6\,\mathrm{s}$ for a single interferogram. Spectra have been generated from raw DC-coupled interferograms using the preprocessor used by the COCCON network, which has been developed in the framework of the ESA project COCCON-

PROCEEDS (Hase et al., 2004; Sha et al., 2019). As suggested by Frey et al. (2015) we use water vapor absorption lines to measure the instrumental line shape (ILS).

The EM27/SUN is mechanically robust but it would not withstand precipitation nor sea spray. For the ship-borne variant, we assembled a small container that houses the EM27/SUN FTS with its solar tracker, a laptop, and several ancillary sensors (GPS, pressure, temperature) similar to Heinle and Chen (2018). During the entire MORE-2 campaign, we placed the container

outside on the port side of the observation deck of the research vessel *Sonne*, which was the uppermost continuously accessible deck available. We chose this spot to avoid obstruction of the direct light path from the sun to the instrument by ship structures. The container is a white lacquered K470 Zarges aluminum box (IP65 waterproof) with dimensions $95\times69\times48\,\mathrm{cm}^3$ and $13.4\,\mathrm{kg}$ mass when empty. Figure 2 shows a photograph of the container deployed on the ship. The solar tracker is a modified version of the custom-built setup used by Klappenbach et al. (2015) consisting of a mirror assembly on two perpendicular rotation

stages that allow for pointing to any azimuth and elevation position of the sun in the overhead sky. For the ship deployment, we covered the solar tracker with a protective housing that has a fused silica wedged window transmitting the incoming sunlight. The solar tracker housing is positioned on top the box and attached to the rotation stages moving with the azimuth and elevation rotations. The precision required for the pointing of the solar tracker is $0.05°$ relative to the center of the sun (Gisi et al., 2011) to keep mole fraction uncertainties due to pointing errors below 0.1%. Our tracking system satisfied this requirement for 79%

of the measurements for which the sun was within the field of view (FOV) of the solar tracker. We observed the largest pointing deviations when high cirrus clouds were present and when the sun was close to the zenith where the azimuth rotation has a singularity. Our filter criteria reliably remove such observations (see section 3.2).

In addition to the main solar tracker, we mounted a f-theta fisheye lens (Fujinon FE185C057HA-1) with a field of view of $185°\times185°$ under a protective acrylic glass dome onto the lid of the box. A camera (IDS UI-3280CP-M-GL Rev.2) observes

the sky through the fisheye lens and provides the position of the sun with an accuracy better than $2°$ while the sun is not within the FOV of the solar tracker. The ambient pressure and temperature sensors as well as the GPS antenna are mounted on the lid as well. The box is equipped with a Pfannenberg PF66000 fan for ventilation on one side and an air outlet on the other side to prevent the box from overheating, both covered by protective lids against sea spray and precipitation. During the whole MORE-2 campaign, the box interior temperature never exceeded $40°\mathrm{C}$. Inside the box, a Raspberry Pi 3 Model

B is the central control unit. It allows for remote access via a network to connect to the laptop controlling the EM27/SUN, an Advantech Ark 2150 embedded PC running the solar tracking software (Klappenbach et al., 2015), and a central storage unit (Synology DS2018 NAS). The Raspberry Pi continuously reads the ancillary sensors for the box-interior temperature, the ambient pressure and temperature, and the GPS position of the instrument. The electronics runs on $24\,\mathrm{V}$ DC provided by an AC C-TEC 2410-10 uninterrupted power supply (UPS). In case of a power cut, the Raspberry Pi securely shuts down all devices

within the approximately $60\,\mathrm{s}$ backup time of the UPS. The whole container weighs about $80\,\mathrm{kg}$ and consumes $350\,\mathrm{W}$ via a regular $230\,\mathrm{V}$ AC line if the electronics is running at full power. Thereby, the ventilation alone takes up $160\,\mathrm{W}$.





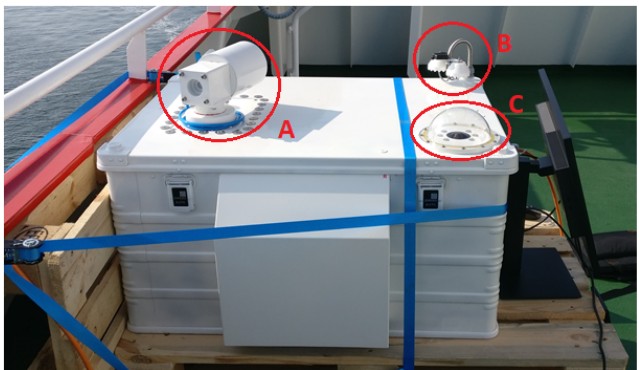 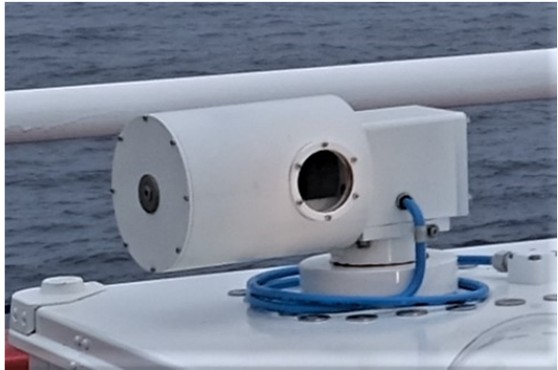

**Figure 2.** Photograph of the instrument container on board research vessel *Sonne* (left) and the solar tracker housing (right). The solar tracker housing (A on the left), the ambient sensors (B on the left), and the fisheye camera (C on the left) are mounted on top of the box. The solar tracker housing (right) consists of a cylinder which rotates around a horizontal axis in elevation direction. The cylinder is mounted on a cube which is able to rotate around a vertical axis in azimuth direction. Sunlight enters the tracker through a fused silica wedged window.

## 3   Data processing

The quality assessment and the retrievals of $XCO_2$, $XCH_4$, and XCO largely follow Klappenbach et al. (2015). Therefore, we summarize the methods here, mostly highlighting the differences and new aspects compared to our precursor study.

### 3.1   Retrieval of $XCO_2$, $XCH_4$, and XCO

The spectral retrieval of the targeted gas concentrations from direct-sun absorption spectra is based on forward modelling of the spectra given a priori concentrations of the molecular absorbers and then iteratively adjusting the concentrations to optimally (in a least squares sense) fit the measured spectra. The spectral retrieval calculates total column number densities of the target gases [GAS] which are a posteriori rationed by the total column number density of oxygen ($[O_2]$) to yield the column-averaged dry-air mole fraction $X_{GAS}$ of the target gas, according to

$$X_{GAS} = \frac{[GAS]}{[O_2]} \cdot 0.2094, \tag{1}$$

where 0.2094 is the constant column-averaged dry-air mole fraction of molecular oxygen. Referencing the target gas column [GAS] to the oxygen column cancels out instrument and retrieval related errors common to both retrievals.

For the spectral retrieval, we use a variant of the RemoTeC algorithm (Butz et al., 2011) which is in use for satellite observation from GOSAT (Butz et al., 2011; Wilzewski et al., 2020), the Orbiting Carbon Observatory (OCO-2) (Wu et al., 2018), and TROPOMI (Hu et al., 2018). We have adapted RemoTeC for transmittance calculations applicable to ground-based direct-sun measurements such as conducted here. Since the spectral resolution of the EM27/SUN is insufficient to extract profile information from the absorption line shapes, RemoTeC retrieves a scaling parameter on the a priori absorber profiles. The spectral



retrieval windows for $CO_2$, $CH_4$, and $O_2$ are located in the SWIR-1 channel and almost identical to the ones used by Klappenbach et al. (2015). The retrieval window for CO is located in the new SWIR-3 channel. Table 1 collects the information on window selection and interfering absorbers. The absorption cross sections of all species are generated from the HITRAN-2016 database (Gordon et al., 2017). Meteorological parameters such as pressure and temperature profiles are taken from the National Centres for Environmental Prediction (NCEP) available at NCEP (2000). These NCEP FNL (Final) Operational Model Global Tropospheric Analyses fields are from the Global Data Assimilation System (GDAS) and have a spatial resolution of $1° \times 1°$ and a temporal resolution of 6 h. The a priori profiles for $CO_2$ and $CH_4$ are taken from CAMS greenhouse gas analysis (Massart et al., 2014, 2016) and for CO from the near real-time operational analysis (Inness et al., 2015, 2019). The profiles are interpolated to the time and location of each individual EM27/SUN measurement. CAMS provides CO profiles with 60 model levels on a $0.4° \times 0.4°$ grid and 6 h temporal resolution for the campaign period in June 2019. The CAMS $CO_2$ and $CH_4$ profiles have 136 model levels on a $0.25° \times 0.25°$ horizontal grid with 6 h temporal resolution.

**Table 1.** Spectral windows with target and interfering absorbers. CIA refers to collision-induced absorption.

| Channel | SWIR-3 | SWIR-1 | | | |
|---|---|---|---|---|---|
| Spectral window / $cm^{-1}$ | 4210-4320 | 5879-6145 | 6173-6276 | 6308-6390 | 7765-8005 |
| Target Absorber | CO | $CH_4$ | $CO_2$ | $CO_2$ | $O_2$ |
| Interfering Absorber | $CH_4$, $H_2O$, HDO, $H_2^{18}O$ | $H_2O$, $CO_2$ | $H_2O$ | $H_2O$ | $H_2O$, $O_2-CIA$ |

## 3.2 Quality filters

The measurements collected during the MORE-2 campaign require quality filtering since cloudy or partially cloudy scenes need to be screened and we want to avoid measurements that are contaminated by the exhaust plume of the ship. To this end, Klappenbach et al. (2015) suggested a cascade of three criteria: a filter based on the DC part of the recorded interferograms, a filter based on the deviation between spectroscopically derived surface pressure and in-situ measured surface pressure, and a filter based on visual identification of steep slopes in the $XCO_2$ timeseries.

During the MORE-2 campaign, our FTS recorded the interferograms with the slowly varying DC part included. The DC part is indicative of the overall incoming radiance and thus, it can be used to track clouds that obstruct the direct-sun view. The DC filter criterion screens measurements either if the DC part $I_{DC}$ is too small to be direct sunlight or if the fluctuation $DC_{fluc}$, defined as

$$DC_{fluc} \equiv \frac{\max(I_{DC})}{\min(I_{DC})} - 1, \tag{2}$$

exceeds 5 %. The DC-filter removes 8.39 % of the data set.

The surface pressure filter compares the in-situ measured surface pressure by the ship's meteorological station and the surface pressure calculated from the spectroscopic measurements as suggested by Wunch et al. (2011). We calculate the spectroscopic pressure from the measured total column number densities of [$O_2$] and [$H_2O$] with





$$p_{dry} = [\mathrm{O_2}] \cdot \frac{M_{\mathrm{O_2}}}{N_A \cdot \xi_{\mathrm{O_2}}} \cdot g \tag{3}$$

$$p_{\mathrm{H_2O}} = [\mathrm{H_2O}] \cdot \frac{M_{\mathrm{H_2O}}}{N_A} \cdot g, \tag{4}$$

where $M_{\mathrm{GAS}}$ is the molar mass of the gas molecule, $N_A$ Avogadro's constant, $\xi_{\mathrm{O_2}} = 0.2314$ the dry air mass mixing ratio of oxygen, and $g$ the gravitational constant.

We scale the spectroscopic pressure to the in-situ pressure with a factor of 0.9693, such that the ratio

$$R_{psf} \equiv 0.9693 \cdot \frac{p_{dry} + p_{\mathrm{H_2O}}}{p_{in-situ}} \tag{5}$$

scatters around unity within the measurement noise under unperturbed conditions. Any measurement for which $R_{psf}$ deviates by more than 0.3 % from unity is excluded from further processing, which leads to a rejection for 6.2 % of the data.

The third quality filter screens the (rare) situation when our EM27/SUN measurements detected the ship exhaust plume. During the MORE-2 campaign, this happened in the morning of June 8, 2019, when the ship's exhaust plume crossed the light
path. The observations show a steep increase of 2 ppm in $\mathrm{XCO_2}$ while $\mathrm{XCH_4}$ and XCO show no increase. We remove 179 measurements (corresponding to 55 min). After applying all filters, a total of 32859 (84.38 %) direct-sun measurements pass the filter process and are considered for further processing.

## 3.3 Bias corrections

After retrieving and quality filtering the $\mathrm{XCO_2}$, $\mathrm{XCH_4}$, and XCO concentrations, the records require bias correction for a
spurious dependency on the solar zenith angle (SZA) causing an artificial diurnal cycle (e.g. Wunch et al., 2011), and for species-dependent scaling factors that adjust our spectroscopic measurements to observations of the TCCON whose stations have been calibrated against standards of the World Meteorology Organization (WMO; Wunch et al., 2010). We find the SZA-dependency for each species from background concentration observations above the Pacific, and the scaling factor to TCCON via the ratio of side-by-side measurements.

The spurious dependency on SZA causes an underestimation of the column-averaged dry-air mole fractions at high SZAs for each of the target species. Wunch et al. (2011) suggested an empirical correction as a function of SZA $\Theta$ according to

$$\mathrm{X}_{\mathrm{corr,GAS}}(\Theta) = \frac{\mathrm{X}_{\mathrm{GAS}}(\Theta)}{\chi_{\mathrm{GAS}}(\Theta)}, \tag{6}$$

where $\mathrm{X}_{\mathrm{corr,GAS}}(\Theta)$ is the corrected column-averaged dry-air mole fraction of the gas under consideration and $\chi_{\mathrm{GAS}}(\Theta)$ is a third-order polynomial of the form

$$\chi_{\mathrm{GAS}}(\Theta) = a\Theta^3 + b\Theta + c, \tag{7}$$





with $a$, $b$, and $c$ free fitting parameters. We perform the fit by splitting our time series in morning and afternoon parts at the lowest SZA of the day and consider only those half-days which contain a measurement at SZA $= (45.0 \pm 0.5)°$, which was the case for 26 half-days. We reference each measurement to the observation closest to SZA $= 45°$, i.e., we choose $\chi_{GAS}(45°) = 1$. Furthermore, we identified half-days which showed actual atmospheric variability by fitting the SZA dependency in a first

attempt and calculating the standard deviation of the fit residuum. We removed a half-day if less than 97% of its observations were within $2\sigma$ of this fit, since this indicates actual atmospheric variability which we do not want to misinterpret as spurious SZA dependency. This results in removing 2, 9, and 9 half-days from the $XCO_2$, $XCH_4$, and XCO records for the fit of the correction polynomial, respectively. Figure 3 shows the corresponding data and the fitted correction polynomials for each target species, and table 2 lists the parameters defined in equ. (7) and the coefficients of determination for each fit. The lowest

coefficient of determination is found for CO, most likely due to the stronger atmospheric variability compared to $CO_2$ and $CH_4$.

**Table 2.** Parameters a, b, and c (and coefficients of determination $R^2$) used for correcting the spurious SZA dependency of the measurements.

| Gas | a / $(°)^{-3}$ | b / $(°)^{-1}$ | c | $R^2$ |
|-----|------|------|------|------|
| $CO_2$ | $-1.91 \cdot 10^{-8}$ | $3.35 \cdot 10^{-6}$ | 1.0015 | 0.857 |
| $CH_4$ | $-2.61 \cdot 10^{-8}$ | $-2.03 \cdot 10^{-5}$ | 1.0032 | 0.928 |
| CO | $-1.74 \cdot 10^{-7}$ | $-5.05 \cdot 10^{-4}$ | 1.0392 | 0.743 |

After correcting the SZA dependency, we adjust our measurements to those of the TCCON station at KIT, Karlsruhe, to ensure traceability to WMO standards (Hase et al., 2015a). To this end, we performed side-by-side measurements at Karlsruhe for a day before (April 30, 2019) and after (July 23, 2019) the ship campaign. Figure 4 shows the TCCON measurements

alongside the EM27/SUN measurements before and after scaling. Following Klappenbach et al. (2015), we calculate the scaling factor as the mean ratio $\gamma_{GAS}$ between the TCCON to the EM27/SUN one-hour means for both days according to

$$\gamma_{GAS} = \left\langle \frac{\langle X_{GAS}^{TCCON} \rangle_h}{\langle X_{GAS}^{EM27} \rangle_h} \right\rangle_{day} , \tag{8}$$

for each of the target species. Table 3 lists the scaling factors $\gamma_{GAS}$ and their error bars, which we calculate as the standard error $\sigma_\gamma$ of the mean using the hourly mean variances $\sigma_{\gamma,h}^2$

$$\sigma_\gamma = \sqrt{\frac{\sum_h^N \sigma_{\gamma,h}^2}{N^2}} \tag{9}$$

$$\left(\frac{\sigma_{\gamma,h}}{\gamma_h}\right)^2 = \left(\frac{\sigma(\langle X_{GAS}^{TCCON} \rangle_h)}{\langle X_{GAS}^{TCCON} \rangle_h}\right)^2 + \left(\frac{\sigma(\langle X_{GAS}^{EM27} \rangle_h)}{\langle X_{GAS}^{EM27} \rangle_h}\right)^2 , \tag{10}$$

where $\sigma()$ is the standard deviation of the hourly mean and $N$ the total number of hours of side-by-side observations. For $CO_2$, the scaling factors are consistent within the error bars, and for $CH_4$ the scaling factors before and after the campaign

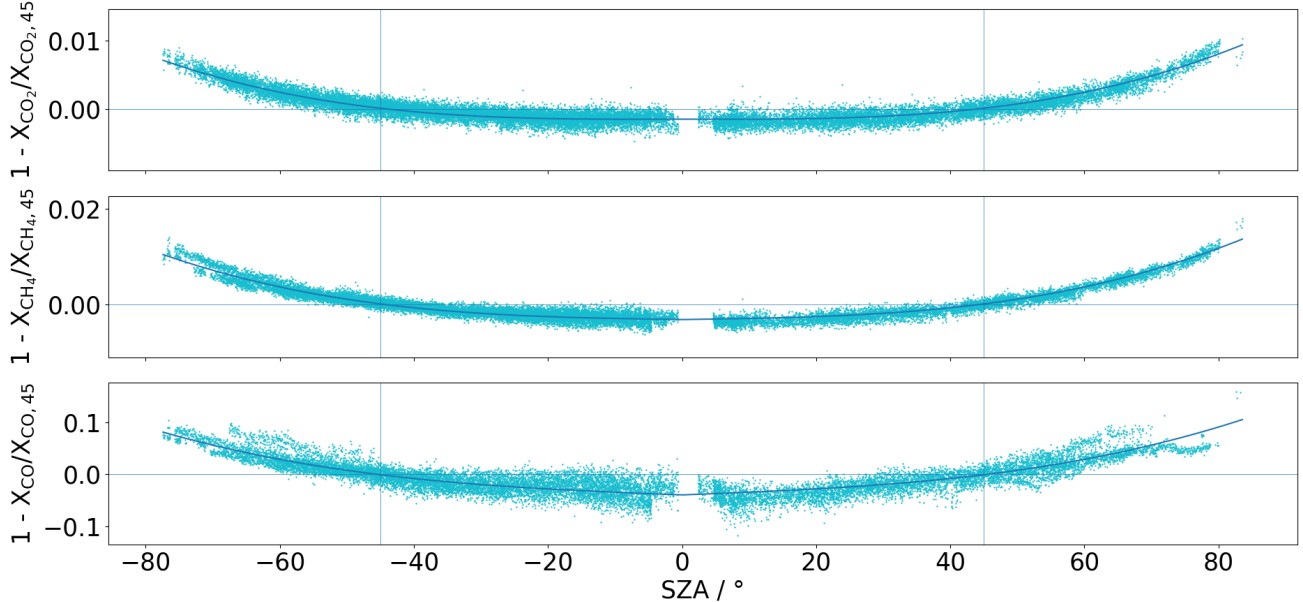

**Figure 3.** SZA dependency of the retrieved $XCO_2$ (upper panel), $XCH_4$ (middle panel), and $XCO$ (lower panel) and the inferred correction polynomial (solid line). Negative SZAs denote morning, positive SZAs afternoon measurements.

are consistent to roughly 3 permil, though the differences are larger than the combined error bars. For CO, the scaling factors
before and after the campaign differ by roughly 2%, which is substantially more than the error bars. We identify a change in the
ILS as the most likely candidate for this difference in the scaling factors. The ILS was measured under laboratory conditions
before and after the campaign. We also performed ILS measurements at the beginning of the campaign on June 2, 2019, yet
it was impossible to assure laboratory conditions there, since the measurements were conducted on deck. The pre-campaign
ILS differs from the one taken on board the *Sonne*, most likely due to rough handling during the transport from Germany to
Vancouver and, in consequence, a slight change in the optical alignment. We could not detect any change of the ILS after
shipment back to Germany from Singapore. Thus, we adjust our measurements with the factors derived from the TCCON
side-by-side measurements on July 23, 2019. Even after applying the scaling factor, Fig. 4 shows that there is some residual
differences between the EM27/SUN and TCCON data growing towards the afternoon. At present the origin of these differences
is unclear. Zhou et al. (2019) suggest further investigations of the TCCON XCO scaling factor based on comparisons of the
TCCON to the Network for the Detection of Atmospheric Composition Change (NDACC) and AirCore Measurements. Should
the TCCON scaling factor be updated in the future, our XCO data will be scaled accordingly.





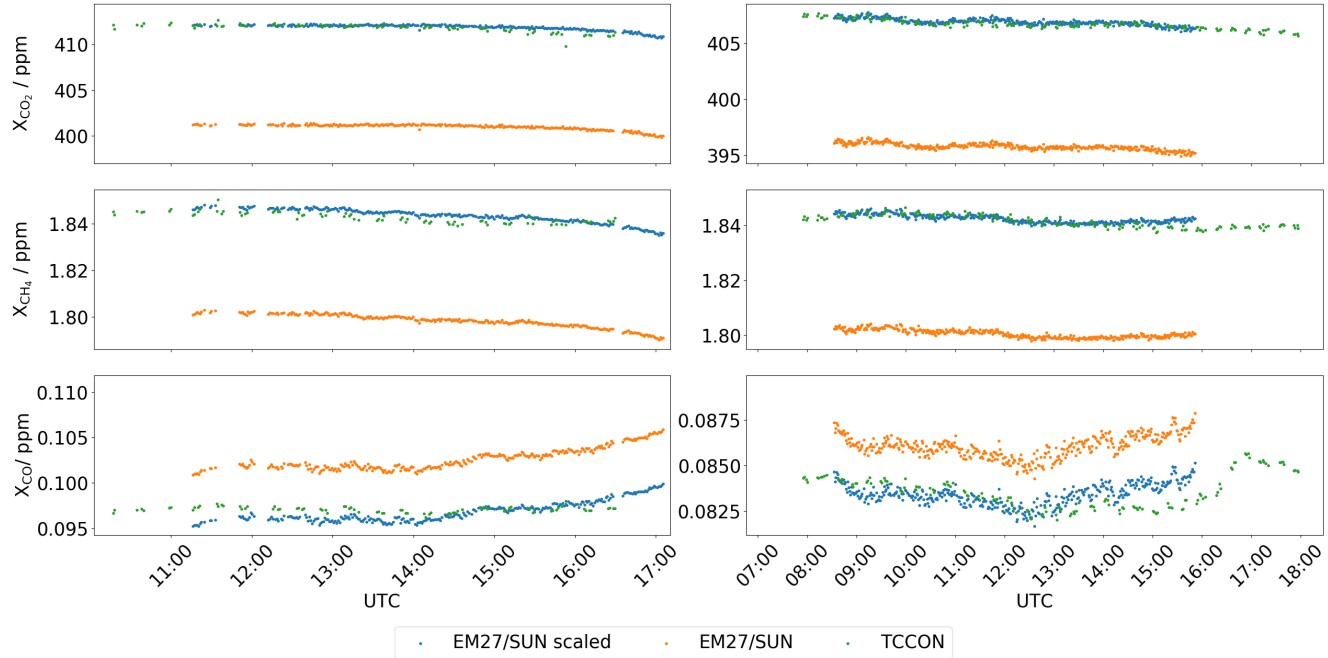

**Figure 4.** Side-by-side measurements of $XCO_2$ (upper panels), $XCH_4$ (middle panels), and $XCO$ (lower panels) by the ship-borne EM27/SUN and the TCCON station at Karlsruhe on April 30, 2019 (left) and July 7, 2019 (right). EM27/SUN measurements before and after scaling are shown in orange and blue, TCCON records are shown in green.

**Table 3.** Scaling factors $\gamma$ for EM27/SUN $XCO_2$, $XCH_4$, and $XCO$ observations as derived from the side-by-side measurements at the TCCON station Karlsruhe (equ. (8)). The uncertainty is the standard error of the mean of the hourly data (equ. (9)).

| Species | 04/30/2019 | 07/23/2019 |
|---------|------------|------------|
| $CO_2$ | $1.0271 \pm 0.0002$ | $1.0272 \pm 0.0002$ |
| $CH_4$ | $1.0251 \pm 0.0003$ | $1.0222 \pm 0.0002$ |
| $CO$ | $0.9436 \pm 0.0018$ | $0.9653 \pm 0.0017$ |

## 4   Comparison to TROPOMI and CAMS

Figure 5 shows the quality-filtered and bias-corrected $XCO_2$, $XCH_4$, and $XCO$ records for our cruise through the Pacific ocean. The trajectory largely follows 30° N of latitude between 140° W and 120° E of longitude crossing the date line on June 14th, 2019. For statistical analysis, we calculate hourly means of our records and use the campaign averaged standard deviation of the hourly means as a measure for our precision, which amounts to $0.24\,\mathrm{ppm}$ ($0.06\,\%$), $1.1\,\mathrm{ppb}$ ($0.06\,\%$), and $0.75\,\mathrm{ppb}$ ($1.03\,\%$) for $XCO_2$, $XCH_4$, and $XCO$, respectively. The data records clearly show that we sampled background airmasses for most of



the time. We calculate a campaign mean and standard deviation throughout the longitudinal section for each species, finding means and standard deviations as little as $411.6 \pm 0.6$ ppm for $XCO_2$, $1835 \pm 7$ ppb for $XCH_4$, and $71 \pm 5$ ppb for XCO.

Figure 5 compares our observations to the CAMS atmospheric composition analyses, the same data from which we interpolated the a priori profiles for our retrievals. We calculate column-averaged dry-air mole fractions from the CAMS vertical profiles. Furthermore, Fig. 5 shows TROPOMI XCO observations for which we apply coincidence criteria of $0.5°$ radius and 4 h time span. The TROPOMI XCO data are available at ESA (2018) and have been retrieved by the SICOR algorithm (Landgraf et al., 2016) which allows for retrievals above clouds. The latter capability is important over the ocean, since the ocean is
dark implying large noise unless the ocean-glint spot is observed or clouds offer a bright reflection target. After filtering with TROPOMI's quality flag (i.e., the internal quality descriptor bounded by 0 and 1 must be larger than 0.5), we find 1783 coincident TROPOMI XCO measurements distributed among 19 days. Although TROPOMI has $XCH_4$ measurement capabilities, we do not discuss these here since there is currently no ocean data available. Likewise, we do not show any OCO-2 or GOSAT data since the number of coincidences was 43 and 9, respectively, and limited to individual days, which we consider too little
statistics for a robust analysis.

Figure 6 depicts the differences between CAMS and our data, and between TROPOMI and our data. We average the differences to CAMS over the entire campaign and calculate the standard deviations of the differences, which amount to $0.52 \pm 0.31$ ppm for $XCO_2$, $0.9 \pm 4.1$ ppb for $XCH_4$, and $3.2 \pm 3.4$ ppb for XCO (see also table 4). Thus, CAMS shows excellent agreement with the ship-borne measurements within $1\sigma$ for $CH_4$ and CO and $2\sigma$ for $CO_2$. Maximum differences
between hourly means of CAMS and EM27/SUN observations amount to $1.0 \pm 0.2$ ppm $XCO_2$, $13.8 \pm 1.3$ ppb $XCH_4$, and $10.3 \pm 0.5$ ppb XCO, where the range is the propagated error using the standard deviations of the hourly mean values. For $XCH_4$, CAMS tentatively shows an underestimation by a few ppb around the date line and an overestimation around $160°$ E. Further, the intra-day variability of $XCH_4$ shows a systematic difference on the order of a few ppb. However, there is no consistent intra-day pattern that fits all the campaign days. Likewise for XCO, there is an intra-day residual pattern on the
order of a few ppb but no consistency that informs on potential model errors or shortcomings of the ship-borne measurements.

TROPOMI XCO also shows very good agreement with our data. The mean difference and standard deviation among the entire campaign record is $2.2 \pm 6.6$ ppb without any systematic pattern correlating with the position in the Pacific ocean. Borsdorff et al. (2019) improved the SICOR/TROPOMI CO data product in comparison to TCCON observations by adjusting the spectroscopic database, decreasing the global mean bias below 1 ppb compared to TCCON station records with a standard
deviation of 2.6 ppb and a TCCON station-to-station bias variation of 1.8 ppb. We investigated whether the small residual differences correlate with the cloud parameters or interfering absorber abundances that SICOR/TROPOMI retrieves simultaneously with XCO. Figure 7 shows the correlations of the differences with the layer height and the scattering optical thickness of the cloud layer as well as the atmospheric methane and water column retrieved by SICOR/TROPOMI. Landgraf et al. (2016) find a retrieval bias in the case of CO enhancements in combination with clouds, which we can not assess from our back-
ground concentration observations. Furthermore, we find no correlation of the XCO differences with the atmospheric methane and water vapor columns retrieved by SICOR/TROPOMI. These two species show spectroscopic interferences with the CO





absorption lines in the SWIR-3 band and thus, they could be candidates for inducing retrieval errors. Overall, our evaluation suggests that SICOR/TROPOMI provides robust results over clouds and for ocean-glint observations.

**Table 4.** Comparison of the EM27/SUN observations to CAMS model data and coincident TROPOMI XCO measurements. The data indicate the mean differences $\pm$ the standard deviation of the differences for the entire record.

| Source | Offset | | |
|---|---|---|---|
| | $CO_2$/ ppm | $CH_4$/ ppb | CO/ ppb |
| CAMS | 0.52$\pm$0.31 | 0.9$\pm$4.1 | 3.2$\pm$3.4 |
| TROPOMI | - | - | 2.2$\pm$6.6 |

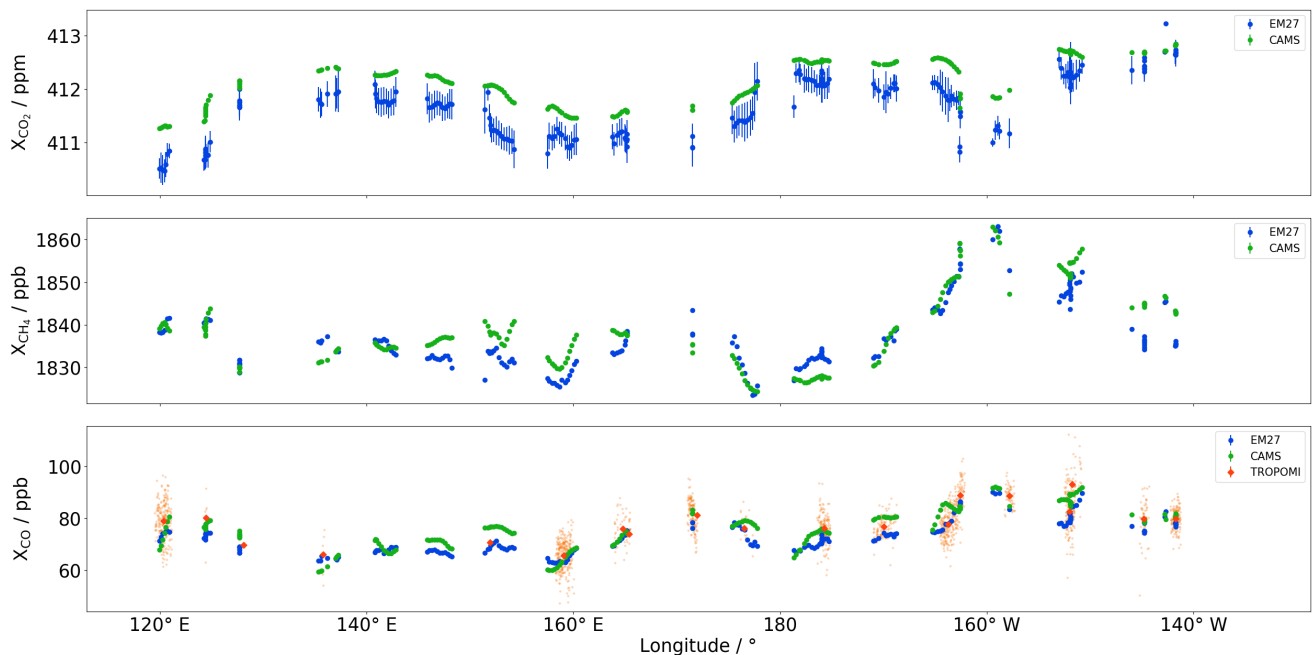

**Figure 5.** $XCO_2$ (upper panel), $XCH_4$ (middle panel), and XCO (lower panel) measured by the ship-gooing EM27/SUN (blue) above the Pacific alongside with coincident CAMS atmospheric composition analysis data (green) and coincident XCO satellite observations by TROPOMI (orange). EM27/SUN measurements and CAMS data are hourly averages while TROPOMI observations are averaged per overflight. Single TROPOMI measurements are marked small in the background.

## 5 Conclusions

We deployed an EM27/SUN FTS on board the German research vessel *Sonne* on the MORE-2 campaign cruise from Vancouver to Singapore leaving port on May 30, 2019 and arriving on July 5, 2019. Compared to our precursor study Klappenbach

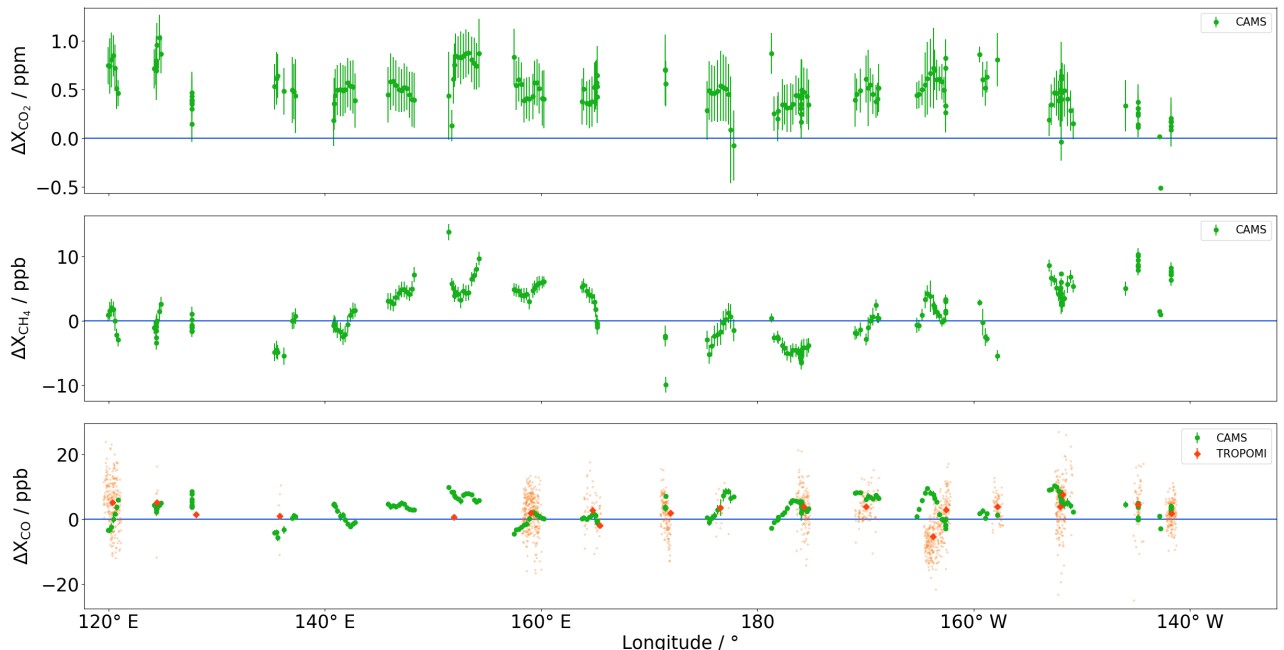

**Figure 6.** Differences between CAMS and our ship-borne EM27/SUN (green) for $XCO_2$ (upper panel), $XCH_4$ (middle panel), and XCO (lower panel), and between TROPOMI XCO and our EM27/SUN (orange). The EM27/SUN measurements were subtracted from the CAMS data and coincident TROPOMI observations.

et al. (2015), our instrument setup was able to withstand environmental conditions and it ran largely without requiring on-site operating personnel. Plus, the sun-viewing FTS was augmented by another detector to collect solar absorption spectra of CO in addition to $CO_2$ and $CH_4$ (Hase et al., 2016). We provide records of the column-averaged dry-air mole fractions $XCO_2$,

$XCH_4$, and XCO for 22 days of measurements on the Pacific ocean largely following $30°$ N of latitude. Our observations are representative of global background conditions; thus, they are useful for assessing the performance of atmospheric models and satellite measurements without perturbations due to local atmospheric variability and they add to the largely land-based validation data provided by the TCCON and the COCCON. Our measurements show an overall precision (hourly standard deviations averaged for the whole campaign) of $0.24\,\mathrm{ppm}$ for $CO_2$, $1.1\,\mathrm{ppb}$ for $CH_4$, and $0.75\,\mathrm{ppb}$ for CO. Systematic errors

due to residual pointing uncertainties, sampling of the ship's exhaust plume, and a spurious dependency on the SZA are treated by filtering flawed data and by empirical corrections. We made our observations compatible with the TCCON through side-by-side measurements before and after the campaign at the TCCON station Karlsruhe. By comparison to our data, we evaluate the performance of the CAMS model for $XCO_2$, $XCH_4$, and XCO and the performance of XCO measurements by the TROPOMI instrument on board the Sentinel-5 Precursor satellite. Averaged over the entire campaign, the differences to CAMS amount to

$0.52 \pm 0.31\,\mathrm{ppm}$ for $XCO_2$, $0.9 \pm 4.1\,\mathrm{ppb}$ for $XCH_4$, and $3.2 \pm 3.4\,\mathrm{ppb}$ for XCO. Furthermore, we find TROPOMI XCO in excellent agreement of $2.2 \pm 6.6\,\mathrm{ppb}$ with the ground-based observations. The instrument is a valuable asset for validation of



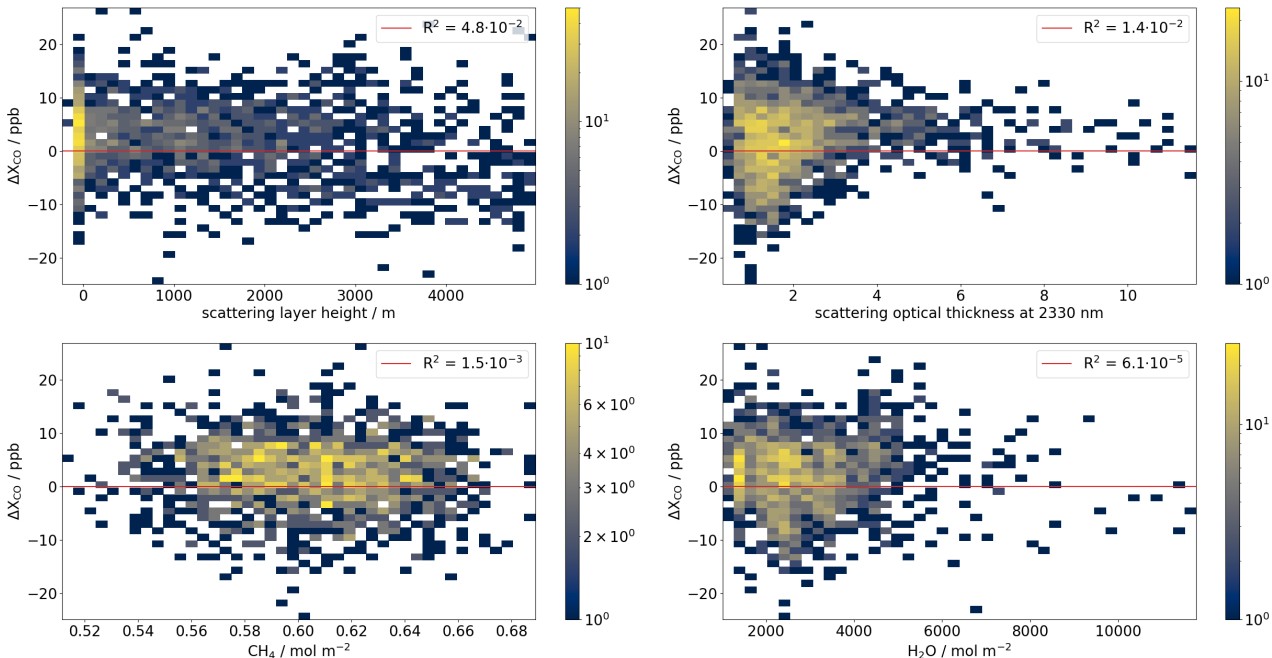

**Figure 7.** Absolute differences between TROPOMI XCO observations and our EM27/SUN measurements plotted as a function of the SICOR/TROPOMI retrieval parameters scattering layer height (upper left), scattering optical thickness (upper right), interfering $CH_4$ (lower left, from weak two-band total column), and interfering $H_2O$ (lower right). The color code indicates (logarithmic) occurrence and the red line is a linear fit with its coefficient of determination $R^2$ given in the upper right of each plot.

satellite observations over sea. In the future, we plan to fully automate our instrument design for routine deployment on ships to enrich validation opportunities over the open oceans where other opportunities are sparse.

## 6 Data availability

The $XCO_2$, $XCH_4$, and XCO records are available for download on PANGAEA at https://doi.org/10.1594/PANGAEA.917240. Preliminary Link: https://doi.pangaea.de/10.1594/PANGAEA.917240 (Knapp et al., 2020).

*Author contributions.* MK and RK developed the ship-borne instrument and operated it during MORE-2. FH contributed heavily on the adjustment process of the EM27/SUN to the TCCON. SK was the chief scientist during MORE-2. AAP, AI, and JB provided the CAMS analyses. JL and TB contributed to the TROPOMI comparison. AB developed the research question. All authors read and provided comments 275 on the paper.



*Competing interests.* The authors declare that they have no conflict of interest.

*Acknowledgements.* We acknowledge funding for the MORE-2 campaign by BMBF (German Federal Ministry of Education and Research) under FKZ 03G0268TD. The development of the COCCON preprocessing tool has been supported by ESA in the framework of the COCCON-PROCEEDS project. The Copernicus Atmosphere Monitoring Service is operated by the European Centre for Medium-Range
Weather Forecasts on behalf of the European Commission as part of the Copernicus programme (http://copernicus.eu).





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
