# Peer review of "Ship-borne measurements of XCO2, XCH4, and XCO above the Pacific Ocean and comparison to CAMS atmospheric analyses and S5P/TROPOMI"

_Earth System Science Data, 2020_

## Referee Comment (RC1) · David Griffith (Referee) · 21 Sep 2020

Knapp et al., ESSD, Ship-borne measurements of XCO2, XCH4, and XCO above the Pacific Ocean and comparison to CAMS atmospheric analyses and S5P/TROPOMI

This paper describes in detail an east-west transect of atmospheric total column amount measurements of CO2, CH4 and CO across the Pacific Ocean near 30°N, from Vancouver to Singapore. The stated purpose of the campaign is to provide an assessment of the potential for ship-borne measurements to validate satellite-based

measurements of greenhouse gases and CO, such as by GOSAT and OCO series satellites, over the oceans. At present such validation relies entirely on land based networks, such as TCCON, so this is a valuable aim, to develop validation methods over the oceans. Although there are few actual coincidences of these campaign measurements with actual satellite measurements, the measurements are compared to two independent datasets, from Copernicus Atmospheric Monitoring Service (CAMS) which assimilates and interpolates data from several satellites for CO2 and CH4, and TROPOMI for CO. The campaign measurements are made by direct solar absorption spectroscopy in the NIR using a Bruker EM27-SUN Fourier Transform spectrometer with custom-built fast response solar tracker suited to shipboard measurements. There is a full description of all components of the measurement system in part referring back to previous work. The instrument, data description and accuracy assessment is quite complete and well suited to ESSD publication. The data are already available in PANGAEA. The paper is acceptable for publication with only a few suggestions for minor revisions to clarify some points.

Technical corrections.

L10: "Precision" is a general term which should not be used for quantitative purposes (see BIPM's Guidelines for Uncertainty in Measurement, GUM). Please specify here in the abstract the measure of precision quoted (0.24 ppm, 1.1 ppb, 0.75 ppb), presumably it is the 1-sigma repeatability of consecutive measurements.

L11: Please add a few words here in the abstract to describe the CAMS product for those readers not familiar with it. In the context it is important to know that this a gridded field of assimilated data from satellites, not a purely model product.

L33: . . . similar column-sensitivity to (not as) the satellites. Also, it appears that there has been no inclusion of averaging kernel information in comparing columns from different instruments and CAMS. This point is not addressed. If the sensitivities are "similar", can you provide a figure for the potential size of the error in ignoring the averaging

kernels?

L45: I suggest replacing "disposes of" with "incorporates"

L82: . . . positioned on top OF the box

L109: ratioed not rationed

L111: TCCON consistently uses 0.2095 for the mole fraction of O2 in air, not the 0.2094 used here. Could you provide a reference to the source of this figure?

L162: Although previously common practice, using the word "calibrated" in comparing TCCON to the SI-traceable scales of the in situ networks is problematic, since many do not consider this to be strictly "calibration". Better to use "validated", or "compared and scaled to " the WMO scales.

L163: the meaning of "background concentration" is not clear here – I think you mean "We determine the SZA dependence for each species from observations over a day in background air when the columns do not vary, and the scaling factors. . .."

Figure 4: It is quite hard to distinguish the blue and green data in these plots, could you choose a more distinct pair, such as blue and magenta?

---

## Referee Comment (RC2) · Anonymous Referee #2 · 29 Sep 2020

Knapp et al's manuscript "Ship-borne measurements of XCO2, XCH4, and XCO above the Pacific Ocean and comparison to CAMS atmospheric analyses and S5P/TROPOMI" is an excellent piece of work describing ship borne measurements of total-column CO2, CO, and CH4 gathered during a transect across the Pacific Ocean at ∼30 deg N. There is a focus on the technical aspects of protecting and successfully deploying a solar-observing Fourier Transform Spectrometer in this challenging environment. There is also a satisfactory description of the trace gas retrieval process and bias correction efforts. Comparisons to both TROPOMI satellite observations, as well

as to CAMS model fields, are also described in detail. As the ocean environment is a measurement poor region for validation of our global models, this work is significant and very worthy of publication following a few minor technical corrections listed below.

The EM27/SUN data is made available by the authors; however, this reviewer struggled to quickly locate and download the exact CAMS product used for both the retrieval a priori and for the final comparison. If this was a special product produced for this campaign then that should be stated clearly and the data should be archived along with the EM27/SUN measurements. Or, if the model fields used are a standard product that I simply overlooked, I suggest including a link to the exact data.

Technical corrections:

L10: is found TO BE 0.24 ppm

L45: Agree with Reviewer #1 – change "disposes of"

L101: I assume the authors are stating that the ventilation takes up 160W when the electronics are running at full power? – this should be made clearer if so.

L147: Is the factor of 0.9693 simply an empirical correction? Is there a physical justification for it?

---

## Author Comment (AC1) · 3 Nov 2020

**Referee Comments - Author Reply**

**Marvin Knapp**

**August 2020**

**1 Response to David Griffith (Referee)**

Dear Prof. Griffith,
we thank you for taking the time to read our manuscript and providing very useful comments. Find our replies below. Referee comments are in italics and the author response is bold.

*L10: "Precision" is a general term which should not be used for quantitative purposes (see BIPM's Guidelines for Uncertainty in Measurement, GUM). Please specify here in the abstract the measure of precision quoted (0.24 ppm, 1.1 ppb, 0.75 ppb), presumably it is the 1-sigma repeatability of consecutive measurements.*
**You are correct, it is the 1-sigma repeatability of the hourly means of hour observations for each species. We changed the formulation in the abstract from "precision" to "[...] 1-sigma repeatability of hourly means [...]"**

*L11: Please add a few words here in the abstract to describe the CAMS product for those readers not familiar with it. In the context it is important to know that this a gridded field of assimilated data from satellites, not a purely model product*
**We changed the formulation to "The Copernicus Atmosphere Monitoring Service (CAMS) models gridded concentration fields of the atmospheric composition using assimilated satellite observations, which show excellent agreement of $0.52 \pm 0.31$ ppm for $XCO_2$, $0.9 \pm 4.1$ ppb for $XCH_4$, and $3.2 \pm 3.4$ ppb for XCO (mean difference $\pm$ standard deviation of differences for entire record) with our observations."**
**to clarify the data product type.**
**Also, we added in L30 "During our campaign in June 2019, CAMS assimilated $XCO_2$ and $XCH_4$ measurements from the Greenhouse gases Observing SATellite (GOSAT) [Kuze et al., 2009], $CH_4$ and CO measurements from the Infrared Atmospheric Sounding Interferometer (IASI) [Crevoisier et al., 2009], and CO measurements from the Measurement of Pollution in the Troposphere (MOPITT) [Drummond**

and Mand, 1996] instrument.".

*L33: [..] similar column-sensitivity to (not as) the satellites. Also, it appears that there has been no inclusion of averaging kernel information in comparing columns from different instruments and CAMS. This point is not addressed. If the sensitivities are "similar", can you provide a figure for the potential size of the error in ignoring the averaging kernels?*

**We changed "as" to "to".**

**The retrieved state vector $\hat{\vec{x}}$ can be written as**

$$\hat{\vec{x}} = \mathbf{A}\vec{x}_{true} + (\mathbf{1} - \mathbf{A})\vec{x}_{ap}$$

**with the averaging kernel matrix A, the unity matrix 1, and the true and a priori atmospheric state vector $\vec{x}_{true}$ and $\vec{x}_{ap}$. We define a total column operator $\vec{h}$ such that $\vec{h}^T \cdot \hat{\vec{x}}$ yields the vertically integrated total column number density of the target gas. Identifying the a priori with the CAMS data, the difference between CAMS and our retrievals (e.g. shown in Fig. 6 of the manuscript) are given by**

$$\vec{h}^T(\hat{\vec{x}} - \vec{x}_{CAMS}) = \vec{h}^T(\mathbf{A}(\vec{x}_{true} + (\mathbf{1} - \mathbf{A})\vec{x}_{ap} - \vec{x}_{CAMS}))$$
$$= \vec{h}^T(\mathbf{A}(\vec{x}_{true} - \vec{x}_{CAMS}))$$

**Therefore, the differences between our retrievals and CAMS have no contribution from the a priori being different from CAMS. Rather, we report the smoothed differences between the "ground truth" and CAMS. The smoothing effect would in theory be accessible by calculating the smoothing error if the covariance of the true state was known. Since the true covariance is very difficult to estimate, we prefer to report the smoothed differences.**

**For the comparisons to TROPOMI CO, the case is different, since TROPOMI uses the TM5 model as an a priori source. In order to quantify the discrepancy introduced by this, we interpolate the TM5 CO volume mixing ratios on our retrieval algorithm vertical grid and to each time and location of our measurements. We calculate the columns in each layer using the airmass from our retrieval and subsequently calculate the a priori contribution to the total column number density $\vec{h}^T(\mathbf{1}-\mathbf{A})\vec{x}_{ap}$ for CAMS and TM5 for each observation. Figure 1 shows the campaign XCO observations using the different a priori datasets, the a priori part of the total column number densities, and their difference. The campaign mean difference caused by the differing a priori profiles is $0.11\pm0.40$ ppb, reaching a maximum of 0.92 ppb. This is a small effect, yet not entirely negligible compared to the small differences we find between our data and TROPOMI CO. We want to address this topic in the paper and reformulated the sentences at at L223 to:**

"Figure 5 compares our observations to column-averaged dry-air mole fractions we calculated from vertical profiles of the CAMS atmospheric composition analyses. These profiles are the same as those we use as a priori for our retrieval. Therefore, the differences between our retrievals and CAMS have no contribution from the a priori being different from CAMS."

Also, we added at L236:

"The SICOR algorithm uses the global chemistry transport model TM5 [Krol et al., 2005] as an a priori source, which introduces a difference in the comparison to our EM27/SUN CO measurements with CAMS a priori [Borsdorff et al., 2014]. We calculate the difference due to the a priori profiles for each EM27/SUN observation and find it to be $0.11 \pm 0.40$ ppb (campaign mean $\pm$ standard deviation) with a maximum of **0.92** ppb. This contribution is small, but not entirely negligible compared to the differences we find between our data and TROPOMI CO."

[Figure]

Figure 1: Top panel: Total column number densities from ground-based observations using CAMS (blue) and TM5 (orange) as a priori input. Middle panel: A priori contribution $\vec{h}^T((\mathbf{1} - \mathbf{A})\vec{x}_{ap})$ from CAMS (blue) and TM5 (orange). Bottom panel: Difference in the a priori contribution to the number density.

*L45: I suggest replacing "disposes of" with "incorporates"*
We changed the formulation according to your suggestion.

*L82: : [...] positioned on top OF the box*
We added the missing word.

*L109: ratioed not rationed*
We replaced the word.

*L111: TCCON consistently uses 0.2095 for the mole fraction of O2 in air, not the 0.2094 used here. Could you provide a reference to the source of this figure?*
**We used the same factor as in our precursor study Klappenbach et al. [2015] to be consistent with them. Since we are scaling our observations to the TCCON data the difference in these factors is without consequence.**

*L162: Although previously common practice, using the word "calibrated" in comparing TCCON to the SI-traceable scales of the in situ networks is problematic, since many do not consider this to be strictly "calibration". Better to use "validated", or "compared and scaled to " the WMO scales.*
**We changed the wording to "compared and scaled to".**

*L163: the meaning of "background concentration" is not clear here – I think you mean "We determine the SZA dependence for each species from observations over a day in background air when the columns do not vary, and the scaling factors [...]."*
**We reformulated the sentence to clarify the matter to "We determine the SZA dependency for each species from observations above the Pacific in background air where the columns are expected to be constant, and the scaling factors [..]"**

*Figure 4: It is quite hard to distinguish the blue and green data in these plots, could you choose a more distinct pair, such as blue and magenta?*
**We changed the green dots to red.**

**2 Response to Anonymous Referee #2**

We thank the reviewer for taking the time to read our manuscript and provide useful comments to it. In the following, referee comments are italic and the author response is bold.

*The EM27/SUN data is made available by the authors; however, this reviewer struggled to quickly locate and download the exact CAMS product used for both the retrieval a priori and for the final comparison. If this was a special product produced for this campaign then that should be stated clearly and the data should be archived along with the EM27/SUN measurements. Or, if the model fields used are a standard product that I simply overlooked, I suggest including a link to the exact data.*
**The CAMS data used in the paper is the official CAMS atmospheric composition analysis. The data for $CO_2$ and $CH_4$ is available via request to Copernicus Service Desk by emailing to copernicus-support@ecmwf.int or via the CAMS enquiry portal in https://atmosphere.copernicus.eu/help-and-support.**

The CO data is available for download at `https://apps.ecmwf.int/datasets/data/cams-nrealtime/levtype=ml/`.
We added the above information to the data availability section.

*L10: is found TO BE 0.24 ppm*
We added the missing words.

*L45: Agree with Reviewer #1 – change "disposes of"*
We changed the formulation according to your suggestion.

*L101: I assume the authors are stating that the ventilation takes up 160W when the electronics are running at full power? – this should be made clearer if so.*
We changed the last two sentences of the paragraph to "The whole container weighs about 80 kg and consumes 190 W via a regular 230 VAC line if the measurement electronics is running at full power. The ventilation consumes an additional 160 W if switched on, which was necessary throughout the campaign."

*L147: Is the factor of 0.9693 simply an empirical correction? Is there a physical justification for it?*
The factor is an empirical correction dealing with an offset between the two different approaches of measuring the surface pressure. A physical reason would be the accuracy of each method, for example due to deficiencies in oxygen spectroscopy.

**References**

T. Borsdorff, O. P. Hasekamp, A. Wassmann, and J. Landgraf. Insights into tikhonov regularization: application to trace gas column retrieval and the efficient calculation of total column averaging kernels. *Atmospheric Measurement Techniques*, 7(2):523–535, 2014. doi: 10.5194/amt-7-523-2014. URL `https://amt.copernicus.org/articles/7/523/2014/`.

C. Crevoisier, D. Nobileau, A. M. Fiore, R. Armante, A. Chédin, and N. A. Scott. Tropospheric methane in the tropics – first year from iasi hyperspectral infrared observations. *Atmospheric Chemistry and Physics*, 9(17):6337–6350, 2009. doi: 10.5194/acp-9-6337-2009. URL `https://acp.copernicus.org/articles/9/6337/2009/`.

James R. Drummond and G. S. Mand. The Measurements of Pollution in the Troposphere (MOPITT) Instrument: Overall Performance and Calibration Requirements. *Journal of Atmospheric and Oceanic Technology*, 13(2): 314–320, 04 1996. ISSN 0739-0572. doi: 10.1175/1520-0426(1996)013⟨0314: TMOPIT⟩2.0.CO;2. URL `https://doi.org/10.1175/1520-0426(1996)013<0314:TMOPIT>2.0.CO;2`.

F. Klappenbach, M. Bertleff, J. Kostinek, F. Hase, T. Blumenstock, A. Agusti-Panareda, M. Razinger, and A. Butz. Accurate mobile remote sensing of $XCO_2$ and $XCH_4$ latitudinal transects from aboard a research vessel. *Atmospheric Measurement Techniques*, 8(12):5023–5038, December 2015. ISSN 1867-8548. doi: 10.5194/amt-8-5023-2015.

M. Krol, S. Houweling, B. Bregman, M. van den Broek, A. Segers, P. van Velthoven, W. Peters, F. Dentener, and P. Bergamaschi. The two-way nested global chemistry-transport zoom model tm5: algorithm and applications. *Atmospheric Chemistry and Physics*, 5(2):417–432, 2005. doi: 10.5194/acp-5-417-2005. URL `https://acp.copernicus.org/articles/5/417/2005/`.

Akihiko Kuze, Hiroshi Suto, Masakatsu Nakajima, and Takashi Hamazaki. Thermal and near infrared sensor for carbon observation Fourier-transform spectrometer on the Greenhouse Gases Observing Satellite for greenhouse gases monitoring. *Applied Optics*, 48(35):6716, December 2009. ISSN 0003-6935, 1539-4522. doi: 10.1364/AO.48.006716.